# Health Literacy in People with Type 1 Diabetes: A Scoping Review

**DOI:** 10.3390/ijerph22060869

**Published:** 2025-05-31

**Authors:** Ilaria Milani, Elisa Cipponeri, Paola Ripa, Arianna Magon, Stefano Terzoni, Silvia Cilluffo, Maura Lusignani, Rosario Caruso

**Affiliations:** 1Department of Biomedicine and Prevention, University of Rome Tor Vergata, 00133 Rome, Italy; ilaria.milani@multimedica.it; 2Department of Biomedical Sciences for Health, University of Milan, 20133 Milan, Italy; elisa.cipponeri@multimedica.it (E.C.); stefano.terzoni@unimi.it (S.T.); silvia.cilluffo@unimi.it (S.C.); maura.lusignani@unimi.it (M.L.); 3Department of Endocrinology, Nutrition and Metabolic Diseases, IRCCS MultiMedica, 20099 Sesto San Giovanni, Italy; 4Nursing School, Ospedale San Giuseppe—MultiMedica, 20123 Milan, Italy; paola.ripa@multimedica.it; 5Health Professions Research and Development Unit, IRCCS Policlinico San Donato, 20097 San Donato Milanese, Italy; arianna.magon@grupposandonato.it

**Keywords:** scoping review, health literacy, Type 1 Diabetes Mellitus, clinical outcomes, nursing, public health

## Abstract

Health literacy (HL) plays a crucial role in the self-management and clinical outcomes of individuals with Type 1 Diabetes Mellitus (T1DM). Despite its significance, research on HL in this population remains fragmented. This scoping review aims to map the existing literature on HL in T1DM and assess its impact on disease management. A systematic search was conducted in PubMed, Embase, Scopus, Web of Science, and Google Scholar, covering studies up to January 2025. The review included 55 studies, primarily focusing on adolescents and young adults, examining key HL dimensions such as numeracy, self-efficacy, nutrition literacy, and the use of digital health technologies. Findings highlight that adequate HL levels are associated with improved diabetes self-care, glycemic control, and quality of life, while barriers such as low numeracy, social anxiety, and socioeconomic disparities hinder disease management. Limited research exists on HL interventions tailored to T1DM, underscoring the need for targeted strategies to enhance patient education and engagement. Given the complex interplay between HL and diabetes outcomes, integrating HL assessments into routine care and developing tailored interventions may improve long-term disease management and quality of life for individuals with T1DM.

## 1. Introduction

Globally, type 1 diabetes (T1DM) affects approximately 8.75 million people, with a growing incidence, as highlighted by the International Diabetes Federation [1]. In 2022, the demographic of individuals over 20 represented 62% of all T1DM cases [2]. In this context, adequate self-care practices to mitigate complications and enhance life quality are key in managing T1DM [3]. Effective self-care behaviors require self-monitoring of blood glucose levels and precise insulin adjustment, contingent upon factors such as diet, physical activity, stress, and coexisting health conditions [4]. However, the intricate balance required for disease management is frequently compromised by socioeconomic disparities, insufficient self-care practices, and a lack of social support networks [5].

As the global incidence of T1DM rises, understanding the pivotal factors underpinning successful disease management and patient outcomes becomes increasingly critical. Among these, Health Literacy (HL) emerges as a cornerstone, extending its influence far beyond self-care practices to encompass clinical outcomes and the overall quality of life for individuals with T1DM [6,7]. HL is defined as the cognitive and social skills that determine the motivation and ability of individuals to access, comprehend, and utilize information in ways that foster and maintain good health; it is categorized into three dimensions: functional HL, encompassing basic reading and arithmetic skills (i.e., numeracy); communicative HL, which involves more complex cognitive and social skills necessary for active participation in healthcare; and critical HL, which enables individuals to critically analyze information and exert greater control over their life circumstances [8]. The dynamic nature of HL necessitates that individuals adeptly navigate the complexities of the healthcare system, making it a crucial area of study for enhancing T1DM self-management and outcomes [9].

The concept of HL is multidimensional, and several tools exist to allow researchers and clinicians to use self-report mode or administrations by an interviewer. The wide range of tools available varies by structure, number of items, management and administration, score, properties, and usage settings. The most widely used tools in people with diabetes, although not specific to the disease, are the Test of Functional Health Literacy in Adults—Short Form, the three brief screening questions, the Rapid Estimate of Adult Literacy in Medicine, and the Health Literacy Survey European Questionnaire [10,11,12]. An additional tool used is the Newest Vital Sign [13]. The short version of the Diabetes Numeracy Test (DNT-15) and the shortened Test of Functional Health Literacy in Adults are tools used in self-report mode [14]; this mode is very practical but requires an adequate cognitive level [13]. Another tool used in the same way is the Health Literacy Scale, which is identified as one of the most appropriate tools for patients with diabetes [14].

While it is well-established that sufficient levels of HL are crucial for effective disease management [15] and could lead to improved behavioral and clinical outcomes [16,17], a nuanced understanding of HL’s impact remains elusive in the context of T1DM [3]. The complexity of T1DM management, which demands a comprehensive grasp of health information and robust communication skills for interacting with healthcare professionals, underscores the need for a detailed exploration of HL [18]. This exploration is vital for enhancing individual patient outcomes and informing clinical practice and healthcare policy at a broader level. The ability to accurately assess an individual’s HL and tailor educational and support interventions accordingly is paramount [9]. In this regard, nurses and healthcare professionals could significantly elevate the level of care provided to individuals, families, and communities, ultimately fostering a more health-literate patient community that is able to address the challenges of chronic disease management effectively.

Notwithstanding the acknowledgment of a positive correlation between HL and diabetes self-care and clinical outcomes [15,17,19,20], existing literature reveals a gap in our understanding of HL’s specific characteristics and its influence on T1DM management [21]. Unlike Type 2 Diabetes (T2DM), where the literature on HL has been extensively synthesized, the research on HL in T1DM remains fragmented and underexplored [22]. This lack of synthesis and consolidation within the T1DM context has impeded the development of targeted interventions that could significantly enhance HL and, by extension, health outcomes for those living with T1DM. In response to the knowledge gap given by the lack of synthesis and consolidation of the literature investigating HL in T1DM, this study aimed to map the literature regarding HL pertaining to T1DM and to elucidate how these characteristics influence health outcomes. Through this review, we seek to answer the following research questions: (RQ1) What are the concepts related to HL in patients with T1DM?; (RQ2) What factors influence HL in patients with T1DM?; (RQ3) What are the relationships between HL and other health-related outcomes in patients with T1DM?; (RQ4) What are the implications for nursing practice among health care professionals who treat people with T1DM?

## 2. Materials and Methods

### 2.1. Design

This research is a scoping review, and its protocol is available elsewhere [23]. According to Arksey and O’Malley, Levac et al., and Peters et al. [24,25,26], scoping reviews are typically applicable to (a) clarify key concepts and definitions in the literature, (b) explore and define knowledge gaps on the subject, (c) provide a comprehensive and up-to-date overview for nurses, researchers, and educators, and (d) facilitate future research and development.

### 2.2. Search Methods: Information Sources and Search Strategy

A systematic literature search was conducted across major biomedical databases: PubMed, Embase, CINAHL, Scopus, Web of Science, and Google Scholar. This search was performed without time restrictions to include the widest possible array of studies, from the earliest available evidence to the most recent findings.

The search strategy is reported in the Appendix A. It was designed to encompass terms related to “Health Literacy” and “Type 1 Diabetes Mellitus”, alongside a broad set of synonyms and related terms to ensure no relevant study was overlooked. The search included studies of any methodological design—quantitative, qualitative, and mixed methods—to provide a holistic view of the field as described in the inclusion/exclusion criteria. This inclusive approach allowed for a comprehensive synthesis of the varied dimensions and domains of HL as they pertain to T1DM.

The developments of the queries, which allowed the reviewers to systematically search the databases, were guided by the Populations, Concept, and Context (PCC) framework, according to the Joanna Briggs Institute (JBI) guidelines [27]. In other words, the queries were developed to be consistent with the review’s questions (i.e., to systematically gather and synthesize evidence on the role of HL in managing T1DM, highlighting areas for future research and potential interventions to improve health outcomes for this population by defining the eligibility criteria within the PCC framework).

Therefore, the literature in this scoping review specifically focused on patients with T1DM. This approach includes patients across all age groups, from children and adolescents to adults, acknowledging the lifelong management required for this condition.

The central concept used to develop the queries was HL. HL encompasses a range of skills and knowledge that enable individuals to make informed decisions about their health. This concept includes understanding medical instructions, navigating the healthcare system, communicating effectively with healthcare providers, and applying health information to improve one’s health status. The review explores how HL influences self-care practices, decision-making, and health outcomes, specifically in the context of T1DM management.

Studies eligible for inclusion could be conducted in various contexts where individuals with T1DM are present. This inclusion strategy includes, but is not limited to, clinical settings (e.g., hospitals, clinics), community settings (e.g., schools, workplaces), and home environments. The review includes studies conducted globally, without restrictions on cultural, subcultural, racial, or gender characteristics, to capture a comprehensive overview of HL in T1DM across different societal and geographical contexts. This approach allows for examining potential geographical specificities and the impact of various healthcare systems on HL and T1DM management.

### 2.3. Inclusion and Exclusion Criteria

This scoping review followed the JBI methodology, which emphasizes the comprehensive mapping of available evidence rather than the critical appraisal or synthesis of effect sizes. Studies were eligible for inclusion if they addressed any dimension of HL in the context of T1DM. Eligible sources included primary research articles, editorials, and conference proceedings that explored HL concepts such as general literacy, numeracy, nutrition literacy, alcohol literacy, and functional or critical HL, as they relate to T1DM self-management, psychosocial well-being, education, or health outcomes. There were no restrictions on publication date, study design, or geographic setting.

Studies were excluded if they focused exclusively on T2DM without reporting disaggregated data for T1DM, or if their primary aim was the psychometric validation of HL instruments without clinical or conceptual application, thus avoiding overlap with “consensus-based standards for the selection of health measurement instruments” (COSMIN) reviews. Studies that did not contribute to the conceptual, contextual, or empirical mapping of HL in relation to T1DM were also excluded. This inclusive and conceptually oriented approach reflects the aim of the review: to identify the scope, thematic domains, and knowledge gaps in the literature on HL in people living with T1DM. Full-text retrieval was supported by the St. Paul’s Hospital—University of Milan library to ensure comprehensive access to relevant studies.

### 2.4. Study Assessment and Eligibility

Following the initial search (i.e., identification phase), the screening of records produced by the search queries was conducted using the online software Rayyan (https://www.rayyan.ai/), a tool designed to streamline the selection process for systematic and scoping reviews [28]. This step was crucial in efficiently managing the literature by eliminating duplicate results and ensuring a focused and relevant pool of studies for further review.

Following the PRISMA Extension for Scoping Reviews (PRISMA-ScR) guidelines [29], the process involved two authors (IM and RC) independently reviewing the titles and abstracts of all studies retrieved. This independent review aimed to assess each study’s relevance and potential contribution to the objectives of this scoping review, as outlined by the PRISMA-ScR checklist and explanation article. Studies that met the inclusion criteria or required further evaluation were reviewed in full text by the same authors (IM and RC). This phase represented the eligibility evaluation that led to including the records that met inclusion/exclusion criteria.

### 2.5. Selection Process Strategy

The selection process followed the PRISMA-ScR guidelines to ensure a transparent and systematic approach. After conducting a comprehensive search across five major biomedical databases, all retrieved records were imported into Rayyan software for screening. First, duplicate records were identified and removed. Two reviewers (IM and RC) independently screened the remaining titles and abstracts based on the predefined inclusion and exclusion criteria. Any discrepancies between reviewers were resolved through discussion or consultation with a third reviewer (EC). Eligible full-text articles were then assessed for final inclusion, considering factors such as relevance to health literacy in T1DM, study design, and population characteristics. Studies focusing exclusively on T2DM, those addressing psychometric validation without direct health literacy analysis, and records without accessible full texts were excluded. The final set of studies was selected for data extraction and synthesis, ensuring a robust and comprehensive mapping of the literature.

### 2.6. Data Items and Data Extraction

Upon the successful completion of the inclusion phase, the data extraction phase was initiated to gather and analyze relevant information from the included studies systematically. A third author (EC) developed a data-charting form to facilitate this process. This form was specifically designed to identify and standardize the variables to be extracted, ensuring a comprehensive and uniform collection of data across studies. The data-charting form enabled the detailed extraction of study characteristics, which encompassed a wide array of variables to provide a thorough overview of each study.

Therefore, the form facilitated the collection of publication details, including the first author, the year of publication, and the country of origin, setting the foundational context of each study. It further delved into study characteristics, capturing the geographical region to understand the environmental context, the country’s economy (to gauge socio-economic influences), the type of record for methodological scrutiny, the journal discipline to appreciate the interdisciplinary perspectives, and the study design, which is crucial for evaluating the methodological rigor. An in-depth examination of each study’s research focus provided clarity on the primary area of investigation, aligning the study objectives with the overarching aim, thereby shedding light on the intended research outcomes. Additionally, details about the population or sample, including demographics and T1DM-specific factors, were gathered to understand the scope and applicability of each study’s findings. Crucial to our comprehensive analysis were the barriers and facilitators identified within each study, which either impede or promote HL and effective T1DM management. Lastly, the key findings and outcomes were extracted and re-elaborated to create a mind map of the main concepts derived from the content analysis of the included literature [30], highlighting the impact on HL and patient care within the T1DM community and following the strategy of synthesis defined in this study. To ensure a robust and unbiased collection of this wealth of information, two reviewers independently extracted and documented the data into a developed Excel spreadsheet, serving as an efficient tool for data analysis.

### 2.7. Synthesis of Results

The results were summarized through a narrative synthesis. This process involved a structured approach to summarizing and interpreting the findings across the collected body of evidence. This process began with a descriptive summary of each study, including its objectives, methodology, population, and main findings, particularly focusing on how HL impacts T1DM management and outcomes. Following this, we identified common themes, patterns, and variations within the data, paying close attention to the different dimensions of HL and the specific barriers and facilitators to effective T1DM management identified across studies. We also considered the geographical, socio-economic, and cultural contexts within which the studies were conducted, as these factors can significantly influence HL levels and diabetes management strategies. This thematic analysis allowed for the identification of gaps in the current literature, emerging trends, and potential areas for future research. A mind map was created from the data extracted from the literature to complement the narrative synthesis with a structured visualization of the results [31]. Two authors then collaboratively discussed and agreed to logically group these themes into broader categories to develop the mind map, ensuring comprehensive data coverage. This grouping was further refined and evaluated through discussions with a third author, bringing additional scrutiny and perspective to the thematic organization.

### 2.8. Risk of Bias

Given the exploratory nature of scoping reviews, which aim to map evidence on a topic and identify main concepts, sources, and research gaps rather than assess the quality of studies, a formal risk of bias assessment process was therefore not implemented. This decision aligns with the guidelines provided by the Joanna Briggs Institute, which suggests that scoping reviews focus on breadth of coverage rather than depth of quality assessment [32].

## 3. Results

### 3.1. Selection Process

The systematic search yielded a total of 1,349 records. After the removal of 154 duplicate reports, 1195 titles and abstracts underwent the screening process. Of these, 1067 records were excluded for various reasons: 989 due to irrelevant outcomes and 78 because they focused on a population different from T1DM. This left 140 records eligible for full-text analysis. Upon further review, 49 full texts were excluded due to not focused on HL, and 24 records were not focused on T1DM. Ultimately, 55 studies met all criteria and were included in the review. The PRISMA flow diagram represents the selection process (Figure 1).

### 3.2. Characteristics of the Included Records

The studies included in this analysis were published between 2005 and 2024 (see Table 1). Most studies (71%) were published between 2015 and 2023, while 29% were from 2005 to 2014. Geographically, the majority of studies were conducted in Europe (39.5%), followed by North America (23.7%), South America (10.5%), Africa (10.5%), Asia (7.9%), and the Middle East (7.9%) (see Figure 2). Regarding economic classification, 71.1% of the studies were conducted in high-income countries, followed by upper-middle-income countries (15.8%) and lower-middle-income countries (13.1%). Regarding publication sources, 97.4% of the included studies were published in non-nursing journals, with only one study appearing in a nursing journal (2.6%). Most (94.7%) were journal articles, while 5.3% were classified as other publication types (e.g., letters, editorials). Observational studies dominated the research landscape, accounting for 73.7% of the studies. Additionally, literature reviews (10.5%), qualitative studies (10.5%), letters (2.6%), and editorials (2.6%) were also included. Appendix A provides a detailed overview of the results according to each research question.

### 3.3. RQ1 Concepts Related to Health Literacy

The research pertinent to our question spans a range of concepts integral to HL in managing T1DM. Among these, nutrition literacy stands out, incorporating knowledge about carbohydrates. Nutrition literacy is the capacity of an individual not only to plan and prepare meals in alignment with dietary recommendations but also to master the confidence and skills necessary for such tasks [43,44,45,46,47]. Another key area is numeracy, or mathematical skills, emphasizing the essential mathematical skills required for engaging with, interpreting, and applying numerical and probabilistic health information toward informed health decisions [33,34,35,36,37,47,48,49,50,69,70,71,72,73,75,76]. Also highlighted is the knowledge pertaining to a healthy lifestyle, notably the role of physical exercise [51]. Alcohol HL is another area of focus [38,39], capturing the individual’s comprehension of alcohol content, their ability to seek out relevant information, and their capability to leverage this knowledge in making health-conscious decisions. Additionally, general literacy encompasses the broader ability to read, understand, and use health information effectively [52,53,54,55,56,70,72,77,78,79,80,81,82,83,84]. These concepts collectively contribute to the multifaceted nature of HL, as delineated by Nutbeam, underscoring its significance in the context of diabetes management [8]. Figure 3 provides a detailed overview of the mentioned concepts.

### 3.4. RQ2 Factors Influence Health Literacy in People with Type 1 Diabetes

#### 3.4.1. Individual Factors: Experience, Education, and Numeracy

The study by Alderson et al. highlighted that experience plays a crucial role in developing the ability to make informed decisions, while active engagement in learning disease control concepts is essential for effective T1DM management [40,57].

Education and numeracy are key determinants of HL. De Morais Borges Marques et al., using the Test of Functional Health Literacy in Adults—Short Form (s-TOFHLA) and Newest Vital Sign (NVS) tools, found that gender, education, income, and anthropometric parameters were not correlated with low HL and numeracy in their studied sample [45]. However, Gomes et al. identified education and age as significant determinants for achieving higher s-TOFHLA scores [7].

In their study, Sancrainte et al. reported that most participants (69.5%) demonstrated adequate HL and numeracy, while 26% exhibited low HL and numeracy levels, with scores increasing with age in youths. However, no correlation was found between HL and numeracy scores and HbA1c levels in caregivers or patients [72].

Muniz et al. demonstrated that inadequate HL is linked to poorer self-care behaviors and suboptimal diabetes management, reinforcing the need for targeted educational interventions. While many individuals with T1DM exhibit insufficient HL, they generally perform better than T2DM patients. Age and years of schooling emerged as key determinants influencing HL proficiency [53].

#### 3.4.2. Cognitive and Functional Impairments

Chaytor et al. further demonstrated that the overall severity of cognitive deficits was independently associated with both diabetes numeracy and Instrumental Activities of Daily Living (IADLs), even after controlling for age, education, frailty, and depression [85].

Similarly, Schwennesen et al. compared HL and self-care behaviors between visually impaired and sighted individuals with T1DM, finding no significant differences in most HL domains, except for finding and understanding health information, which was significantly better in the sighted group. However, self-care behaviors were comparable between the two groups, suggesting that visually impaired individuals may develop compensatory strategies for diabetes self-management [55].

#### 3.4.3. Social and Psychological Factors

High disease burden has been associated with social withdrawal, a factor that may influence individual behaviors [43]. However, peer support, easy access to information, membership in patient associations, and effective therapeutic relationships have been shown to enhance HL levels [54,58].

Wagner and Olesen further emphasized that functional HL and social support play a significant role in how education influences disease control, as assessed using the HLQ instrument (Danish version) and a survey of educational attainment [56].

Moreover, HL training has been shown to enhance self-efficacy and reduce social anxiety among adolescents with T1DM, equipping them with the necessary skills to navigate social interactions and manage their condition with greater confidence [77].

In line with this, Rahimi et al. identified an inverse and significant relationship between HL and social anxiety, as well as a direct positive relationship between HL and self-efficacy, further emphasizing the crucial role of HL in both psychological well-being and disease management [82].

A small subset of studies addressed the role of caregivers and family members, particularly in pediatric and adolescent T1DM populations. Moosa et al. found that both patients and their primary caregivers exhibited poor performance on diabetes-related mathematical tasks, underscoring shared gaps in numeracy that may hinder effective disease management [36]. Similarly, Owusu et al. reported that caregivers demonstrated knowledge and skills in blood glucose monitoring and hyperglycemia management, although gaps remained in areas like carbohydrate counting [54]. Lappenschaar et al. included parents of children with T1DM, reinforcing the relevance of caregiver involvement in carbohydrate-related literacy [47]. These findings highlight the potential influence of caregiver HL on patient outcomes and suggest a need for more research in this domain.

#### 3.4.4. Disease Management and Dietary Knowledge

Counting carbohydrates and understanding proper dietary choices plays a crucial role in dietary decision-making [38,39,40,43,47]. Furthermore, higher scores on the Adult Carb Quiz were associated with better glycemic control, confirming the importance of carbohydrate counting education in diabetes management. As highlighted in the editorial by Briggs Early K., repeated education sessions, rather than a single intervention, were associated with higher knowledge retention, reinforcing the need for continuous learning to support dietary self-management [69].

However, several barriers hinder HL development, including high food costs and limited dietary knowledge [57]. Arghittu et al. observed that food literacy levels were generally higher among women but declined with age, suggesting the need for sustained nutritional education across the lifespan [44].

#### 3.4.5. Socio-Demographic Disparities and Comorbidities

The presence of comorbidities such as dyslipidemia, hypertension, diabetic retinopathy, and socio-demographic characteristics related to occupational status and marital status have been identified as risk factors for lower adherence [86].

Research by Zuercher et al. found that low functional HL (measured using screening questions for limited HL) was associated with low individual income and long diabetes duration [51]. Similarly, the narrative review by Gandhi et al. identified ethnicity and poor language skills as factors influencing HL, with Hispanic and Asian individuals showing lower HL levels compared to White individuals in the United States [59]. Further supporting this, Hillson R., in an editorial, linked low HL to poorer diabetes knowledge but found insufficient evidence of its direct impact on diabetes care processes or outcomes. Among adults with T1DM, poor literacy and numeracy skills were associated with higher HbA1c levels, particularly in those with low numeracy, even after controlling for confounding factors [81].

Additionally, studies by Olesen et al. and Piatt, using the Danish Version of the Health Literacy Questionnaire (HLQ) and the Newest Vital Sign (NVS) tools, reinforced the findings of De Morais Borges Marques et al., confirming that education remains a crucial factor in HL development [18,41].

#### 3.4.6. Age-Stratified Analysis of Health Literacy, Numeracy, and Diabetes Management

Age was recorded in years and categorized into four groups for analysis, following the stratification approach of Berens et al. [87].

Under 15 years (Children and Adolescents): In this age group, both general health literacy (HL) and numeracy were analyzed. Younger age correlated with lower HbA1c, while other factors showed no association. Poor math performance highlighted educational gaps, with low numeracy negatively impacting diabetes management through its inverse correlation with A1c levels [36,70].

15–29 years (Adolescents and Young Adults): In this age group, literature highlights nutrition literacy, alcohol use, general HL, self-efficacy, social anxiety, and numeracy. Diabetes management often leads to social withdrawal due to food concerns, driven more by disease complexity than a lack of dietary knowledge [43]. Findings on young adults’ food literacy are conflicting, with some studies reporting adequacy [46], while others highlight poor health and nutrition literacy, showing no links to gender, education, income, or anthropometric variables [45]. Although many individuals demonstrate adequate HL and numeracy, a substantial portion struggles, with scores improving with age but showing no correlation with HbA1c in patients or caregivers [72]. However, peer support and trusted healthcare relationships enhanced HL and empowerment [58]. Alcohol knowledge was particularly poor [38,39]. Young individuals exhibited strong self-monitoring and hyperglycemia management skills [54] with higher HL linked to greater self-efficacy and lower social anxiety [82]. However, long-term data from Röhling showed no significant HL changes over 12 years in students [83].

30–45 years (Adults): In this age group, key areas include HL, adherence, glycemic control, numeracy, diabetes knowledge, and complications. Most individuals with T1DM have adequate HL but experience high diabetes distress [86]. HL and QoL are correlated with diabetes complications [60]. In Gomes’ study, only 18% of individuals with T1DM had inadequate HL, and disease knowledge was often poor [52]. Carbohydrate knowledge moderately correlated with HbA1c [47], while low numeracy was linked to poorer glycemic control [34,35].

46–64 years (Middle-aged Adults): Bouclaous and Piatt reported low HL levels in most participants [41,49]. Diabetes self-efficacy correlates with lower A1c [71], while numeracy predicts adherence but not glycemic control [76]. Broos et al. found no association between HL, diabetes knowledge, and glycemic control after one year of CGM use. However, HL influences glycemic control, severe hypoglycemia, hypoglycemic coma, and absenteeism in T1DM [16,61,88]. CGM improves clinical outcomes, particularly by reducing A1c, yet many insulin pump users struggle with diabetes numeracy [73,79]. In this population, evaluating health information predicts diabetes management independently of education or disease duration. Higher HL is linked to lower HbA1c and reduced variability [18,62].

65+ years (Older Adults): Chaytor and Zuercher found that older adults with T1DM struggle with numeracy and functional HL, limiting their ability to manage diabetes independently. Cognitive decline and low HL levels hinder insulin dosing accuracy, glycemic trend interpretation, and the effective use of digital self-management tools [51,85].

### 3.5. RQ3 Relationships Between HL and Health Outcomes

#### 3.5.1. Impact of HL on Glycemic Control and Self-Management

Several studies have examined the relationship between health literacy, numeracy, and glycemic control. Elevated disease burden has been associated with increased hemoglobin A1c (HbA1c) levels [43].

Despite the use of various HL and numeracy assessment tools such as Rapid Estimate of Adult Literacy in Medicine (REALM), the math section of the Wide Range Achievement Test (WRAT-3R), Diabetes Self-Management Questionnaire (DSMQ), and the Arabic version of the Michigan Diabetes Numeracy Test (DNT-15), findings suggest that good numeracy skills promote effective self-management behaviors and improved HbA1c control [3,33,48,71].

However, Chima et al. found that numeracy alone was not sufficient to enhance individual engagement in diabetes management. Instead, the broader concept of HL, which includes both cognitive and functional skills, played a more significant role in effective disease management [63]. Similarly, Moosa and Sagal, using the Diabetes Mathematical Questionnaire (DMQ), found that poor numeracy levels negatively impact disease management, as well as an individual’s economic, social, and psychological well-being [36]. Further supporting this, Marden et al. found that numerical skills were inversely correlated with glycemic control, while general literacy showed no significant association with HbA1c levels [34]. Two years later, the same authors, using The Skills for Life Initial Assessments tool on the same cohort, confirmed that individuals with higher numerical skills exhibited better HbA1c levels [35]. Moreover, poor numeracy levels have been linked to negative economic, social, and psychological conditions, as well as worse clinical outcomes [35,50]. Similarly, Zaugg et al. reported that diabetes numeracy was a predictor of adherence to care but not of glucose control, further underscoring the complex relationship between numeracy skills, self-management behaviors, and metabolic outcomes [76]. Additionally, Gomes et al., using the s-TOFHLA, found that individuals with well-controlled hemoglobin A1c levels demonstrated adequate HL, underscoring the link between health literacy and effective diabetes management [7]. Individuals with T1DM have demonstrated better levels of numeracy and HL as measured through s-TOFHLA, DNT-15, and The Single Item Literacy Screener tools compared to those with T2DM [49,86]. Leduc et al. found that younger age predicted lower HbA1c levels, but gender, diabetes duration, numeracy, mother’s education, and self-management scores were not significant predictors [70].

Beyond numeracy, food and alcohol literacy also play a role in diabetes management. Itzkcovitz’s study, using the Short Food Literacy tool, found that individuals with T1DM had greater food literacy compared to the control group [46]. Additionally, Barnard et al., utilizing two versions (AUDIT-C and AUDIT-3) of the Alcohol Use Disorders Identification Test (AUDIT), identified low alcohol literacy levels among the study population, further emphasizing the importance of health literacy in lifestyle-related diabetes risk factors [38,39].

#### 3.5.2. The Role of HL in Disease Outcomes and Technology Use

Studies exploring the effect of HL on diabetes management outcomes have produced mixed results. Broos et al. found no association between adequate HL (measured using NVS and Patient Education and Knowledge (PEAK)) and glycemic control in patients using continuous glucose monitoring (CGM) [61]. However, in a follow-up study, the same authors reported that higher HL levels in CGM users were linked to a lower risk of short-term negative clinical outcomes and an improved quality of life, as reflected by reduced absenteeism from work and school [16].

Similarly, Turrin and Trujillo found that many patients undergoing insulin pump therapy faced challenges related to diabetes numeracy, particularly older individuals and those with higher A1c levels. Notably, a portion of the patient population demonstrated limited numeracy skills, which could significantly hinder their ability to effectively manage insulin pump therapy [73]. In an editorial, Seger emphasized that effective T1DM management requires advanced technologies, medications, and treatments, all of which depend on adequate HL. Low HL has been associated with poorer health outcomes, reduced adherence, and challenges in navigating healthcare, particularly in individuals with impairments or limited technological skills. Screening tools can help identify low-HL patients and facilitate targeted interventions to enhance self-management and clinical outcomes [84].

#### 3.5.3. Disparities in HL and Disease Outcomes

Zuercher et al. analyzed the link between Functional Health Literacy (FHL) and diabetes care processes, reporting that low FHL correlated with lower income, longer diabetes duration, and reduced self-efficacy, but showed no direct association with care processes [51]. Findings from Cheng et al. further emphasized that poor diabetes-related knowledge was strongly associated with low educational levels and older age. Their study identified a significant relationship between inadequate diabetes knowledge and poor glycemic control, as well as negative impacts on adherence to self-management practices, including medication compliance and blood glucose monitoring [89]

#### 3.5.4. HL and Clinical Outcomes

Drown et al. found that low HL was linked to greater challenges in disease management, resulting in negative clinical outcomes. Similarly, Esen and Aktürk Esen reported that limited HL hindered access to healthcare services, significantly reducing individuals’ quality of life [6,57].

In contrast, an observational follow-up study by Röhling et al. found no significant changes in HL levels related to diabetes or other cardiovascular risk factors among adolescent cohorts over a 12-year period. These findings suggest that, in certain populations, HL levels may remain stable over time, even with prolonged exposure to risk factors [83].

#### 3.5.5. HL, Perceived Competence, and Adherence

The association between perceived HL and actual self-management behaviors has also been examined. Kane et al. found that adequate HL levels improved disease management in adolescents [64]. Several studies have highlighted how adequate HL levels have a positive impact on glycemic control and are positively correlated with good disease management, resulting in positive health outcomes [3,18,42,65].

However, these results emerged only in studies that measured the perceived (subjective) concept of HL [65], and Reagan et al.’s study did not show statistically significant results following multiple linear regression [62].

Additionally, De Carvalho et al. used the Problem Area In Diabetes (PAID) and Brief Medication Questionnaire (BMQ) to assess adherence, revealing high scores indicative of poor therapeutic adherence among study participants [86].

### 3.6. RQ4 Implications for Nursing Practice

The assessment of HL and numeracy levels has proven to be important in effectively educating individuals in disease management [18,37,41]. Continuous education and easy access to information have been shown to facilitate the development of HL [54,57]. In their study conducted in Pakistan, Mangi et al. emphasize that low levels of disease-related knowledge are secondary to poor education, and the insufficient level of knowledge, measured by the Knowledge Attitude and Practice (KAP) instrument, revealed in the study was also due to the absence of services for managing individuals with the condition [52]. Zaugg et al. further demonstrated that patients treated by diabetes specialists exhibited higher numeracy, education levels, and completion of diabetes education compared to those managed by primary care physicians. However, these advantages did not translate into improved A1c levels, suggesting that while specialized care enhances knowledge and numeracy, additional factors may influence metabolic control [76].

### 3.7. Main Concepts: Mind Map of the Results

As shown in Table 2, based on the extracted themes from the included articles [3,6,7,16,18,33,34,35,36,37,38,39,40,41,42,43,44,45,46,47,48,49,50,51,52,53,54,55,56,57,58,59,61,62,63,64,65,66,69,70,71,72,73,75,76,77,78,79,81,82,83,84,85,86,88,89], a comprehensive mind map depicted in Figure 4 delineates the multifaceted relationship between HL and the management of T1DM [40,45,48,53,54,55,56,57,64,72,81,85,86,89], encapsulating essential elements such as experience and active learning [40,57], the impact of socialization and peer support [43,56,58,77,82], dietary knowledge and carbohydrate counting [43,44,45,46,47,54,57,69], and the significant barriers posed by economic and educational limitations [52,57,67].

It further explores the critical role of clinical outcomes and HL [3,7,33,34,35,36,41,42,45,47,48,49,50,56,59,62,63,64,65,71,75,76,78,83,86], emphasizing the nuanced perspectives on digital health technologies [16,47,55,58,61,73,84], the influence of HL over the disease’s duration [18,51,57,70,83], and the specific challenges surrounding alcohol literacy [38,39] and therapeutic adherence [63,86].

Additionally, the diagram highlights the relationships between HL and health outcomes [3,6,37,57,59,64,65,71], underscoring the importance of good numeracy for effective self-management [3,33,34,35,36,37,47,48,49,71] and the positive implications of adequate HL in mitigating negative clinical outcomes [7,35,37,46,62,64,65,71]. Lastly, the mind map presents the implications for nursing practice [57,66], stressing the vital role of education in disease management [18,35,44,54,56,57], the necessity for a comprehensive care approach, sensitivity towards socioeconomic and cultural factors [34,35,56,59], the potential benefits of leveraging technology [16,58,61], and the importance of addressing barriers to HL, offering a holistic view of the intersections between HL, patient outcomes, and clinical practice in the context of T1DM.

## 4. Discussion

This scoping review synthesizes two decades of research on HL in T1DM, drawing primarily from studies conducted in high-income countries and published in medical journals. A notable diversity in study designs is observed, with a predominance of observational studies and two qualitative studies identified within nursing journals [54,89], underscoring the interdisciplinary nature of HL research and the nascent but still limited interest in nursing research in this regard. Most studies include a sample aged 15 to 29 years, focusing on key areas such as nutrition, self-efficacy, social anxiety, HL, distal technologies, and numeracy.

The body of work reviewed reveals a consistent focus on individuals diagnosed with T1DM, providing valuable insights into this population’s specific challenges and needs. A significant portion of the literature examines numeracy, uncovering its critical role in disease management [3,35,36,49,50,71,73]. Contrasting opinions emerge on the association between high numeracy and better hemoglobin A1c levels [34,35,47,70,72,76]. Despite mixed evidence, studies consistently show that strong numeracy skills improve self-management behaviors [3,33,48,71,76]. Chaytor et al. identified three major barriers to effective numeracy in diabetes care: older age, lower education levels, and increased depressive symptoms. These factors further complicate disease self-management, emphasizing the need for personalized interventions [85]. Despite this, the relationship between numeracy and patient engagement remains unclear. Studies report conflicting results regarding its influence on glycemic control and self-care behaviors [36,63]. Kerr found that education programs targeting numeracy and HL can improve self-efficacy and glycemic control in adults with diabetes. However, these benefits often decline over time, highlighting the need for ongoing, adaptive education programs to sustain long-term engagement [75].

Adequate levels of HL are correlated with better health outcomes (i.e., lower A1c) [7,56,64]. In this context, Hillson cautions that while this correlation exists, some studies fail to establish a direct causal link between HL and glycemic control. This suggests that additional factors—such as numeracy, socio-economic status, and access to diabetes education—may play a more significant role in metabolic outcomes [81]. Cheng et al. highlight that individuals with higher socio-economic status and better diabetes-related knowledge achieve better glycemic control due to improved access to healthcare resources and greater adherence to self-management practices. This underscores the importance of targeted education and economic support for better health outcomes [89]. Marciano et al. also found that adequate HL correlates with improved health outcomes. However, these results were observed mainly in studies measuring perceived HL rather than objective literacy levels. This discrepancy suggests a need for further research to clarify HL’s role in diabetes management [65]. This discrepancy calls for further research to unravel the nuances of HL’s role in diabetes care.

In examining the mechanisms through which HL influences diabetes management, this review identified several recurring pathways. First, HL contributes to improved understanding of disease physiology and self-care routines, enabling individuals to make more informed decisions regarding carbohydrate counting, insulin titration, and dietary adjustments. Second, HL enhances functional engagement with diabetes-related technologies, such as continuous glucose monitors and insulin pumps, facilitating more accurate data interpretation and responsive self-management. Third, HL was shown to bolster psychological competencies, including self-efficacy, confidence, and reduced social anxiety, which in turn support sustained engagement in treatment behaviors. Lastly, HL interacts with structural enablers, such as educational access and socioeconomic resources, amplifying or mitigating its impact on outcomes depending on contextual support. These findings suggest that HL operates through cognitive, behavioral, emotional, and systemic pathways, reinforcing its multidimensional role in chronic disease management.

In addition, while this review captures a broad range of studies across different age groups and geographic regions, variations in HL must be interpreted with caution. As described in previous literature, adolescents, young adults, and older individuals often face distinct health system challenges, educational exposures, and developmental trajectories that shape their health literacy differently [74,90,91]. Likewise, disparities across geographic and cultural contexts, including healthcare infrastructure, socioeconomic status, and education systems, may contribute to divergent patterns in HL and diabetes management [92,93,94].

The impact of comorbidities, disease duration, and socio-demographic factors on HL levels and treatment adherence is widely recognized. Low HL levels are often linked with chronic illness and increasing susceptibility to health issues [18,41,42,45,51,59,86]. Newly diagnosed individuals often struggle with understanding self-management practices, particularly when literacy levels are low [57]. Moreover, those with limited HL and restricted access to healthcare services often experience a lower quality of life and an increased risk of negative clinical outcomes [6].

According to this scoping review, T1DM significantly impacts the psychosocial and economic well-being of individuals and their families. The heavy burden of the disease is often linked to social withdrawal, which can shape behavioral patterns [43]. Training in HL has been shown to boost self-efficacy and alleviate social anxiety in adolescents with T1DM, equipping them with essential skills to navigate social interactions and manage their condition more confidently [77]. Consistently, Rahimi et al. found that higher HL levels correlate with reduced social anxiety and increased self-efficacy. These results underscore the vital role of HL in both mental health and effective disease management, emphasizing the need for interventions that foster HL to build confidence and ease psychological distress [82]. The negative impact of T1DM on social engagement and education highlights the necessity of support systems that minimize these challenges and strengthen HL development [43,56]. Key factors for effective disease management and improved HL include education, healthcare access, peer support, patient associations, and active participation. Personal experiences with diabetes also contribute to informed health decisions, a pattern observed even among children [40,57].

Enhancing HL and mastering carbohydrate-counting skills have proven effective in promoting adherence to low-sugar diets and improving A1C levels [43,47,56]. Notably, high A1C levels are not always linked to poor dietary knowledge. Instead, they often stem from the emotional burden of managing diabetes, which can lead to social withdrawal [43]. Research suggests that social support plays a key role in glycemic control and may influence an individual’s HL [56]. However, high A1C levels are not necessarily linked to poor dietary knowledge. Instead, they are often driven by the emotional burden of diabetes management, which can lead to social withdrawal [43]. Given these findings, targeted interventions to improve HL and strengthen social networks could help reduce social disparities and enhance overall health outcomes. Additionally, research indicates gender differences in dietary knowledge, with women tending to have greater knowledge than men, although this knowledge declines with age [44]. Among young adult populations, studies have reported good glycemic monitoring skills, complication management, self-confidence, and engagement [46,54]. However, some authors argue that gender, education level, economic conditions, and anthropometric variables are not directly correlated with literacy levels [45,57,70], yet the relationship with education presents a complex picture [18,41,53,62,67]. Rather, poor literacy and HL levels are responsible for poor glycemic control and difficulties in disease management [45,57,89]. Limited data are available regarding carbohydrate counting skills, crisis management during hypoglycemic episodes, and disease management during life stages such as pregnancy [54].

Interestingly, no associations have been observed in the literature between education level and HL and significant changes in glycemic variability indices in individuals using glucose monitoring technology. Education level and HL were measured using the PEAK and NVS-D tools. From the results obtained, the median scores indicated high levels of education and HL in the studied sample. Therefore, it could be inferred that no associations between diabetes and HL were observed due to the high basic education level.

However, adequate HL levels positively impact glycemic control, reducing the risk of complications and missed school or work days among individuals using continuous glucose monitoring systems [16]. To maximize the benefits of emerging technologies, it is essential to foster HL and patient empowerment through person-centered care [58]. While technology plays a critical role in diabetes self-management, access to these tools remains a challenge for certain patient groups. Schwennesen et al. highlight the need for educational, informational, and technological advancements to support individuals with T1DM and visual impairments. Additionally, healthcare professionals should receive specialized training to help these patients access non-visual self-monitoring tools, which are essential for effective disease management [55]. Turrin and Trujillo further emphasized that many patients undergoing insulin pump therapy face challenges related to diabetes numeracy, particularly older individuals and those with higher A1c levels. Limited numeracy skills could significantly hinder their ability to effectively manage insulin pump therapy, underscoring the need for targeted education and support strategies [73]. Technology has also been proposed as a tool to assist with carbohydrate counting. However, its effectiveness in improving accuracy remains limited [69]. These findings reinforce the importance of personalized, accessible interventions that address numeracy skills, HL, technological barriers, and the unique needs of vulnerable groups to improve autonomy and health outcomes.

Zaugg et al. found that patients treated by diabetes specialists demonstrated higher numeracy, education levels, and completion rates of diabetes education programs compared to those managed by primary care physicians [76]. However, despite these advantages, disparities in HL and access to care remain significant barriers to effective disease management. Factors such as the high cost of food, limited dietary knowledge, and distance from health facilities pose substantial challenges to developing HL, and the absence of dedicated services for managing individuals with T1DM is one of the reasons for insufficient knowledge regarding it [52,57]. The barriers identified in the study by Drown et al. are contextualized in a low-income country like Malawi, where the disease significantly impacts psychosocial and economic levels. Considering the average age of the participants in the study (31.8 years), it is interesting to consider the possibility of intervening with the population through the use of care transition models to support young adults in effectively managing the condition. Such models involve the support of multidisciplinary teams, preparation for transition, and the use of telemedicine [95,96,97,98]. This approach falls within the framework of integrated disease management models, where the individual is at the center of a comprehensive territorial network [99].

The literature examined on this matter has not proven to be extensive. Studies conducted by Barnard et al. highlight the limited knowledge among young adults with T1DM regarding alcohol and its sugar content [38,39]. This finding draws the attention of healthcare providers, patient associations, and caregivers to raise awareness among individuals affected by the condition about responsible alcohol consumption.

Two reports addressed therapeutic adherence, highlighting that adequate levels of HL are associated with higher medication engagement. However, this relationship did not show statistically significant results concerning numeracy [68,89]. Kerr emphasized that low numeracy in patients complicates risk perception, screening, and medication adherence, potentially undermining disease management. While HL appears to play a crucial role in promoting adherence, limited numeracy skills may still present a barrier to effective self-care, particularly when understanding dosages, risk factors, and the need for routine screenings [75]. These findings suggest that interventions aimed at improving medication adherence should focus on HL and address numeracy skills to enhance patient comprehension and engagement in their treatment plans.

Our scoping review has several limitations. We only included papers in English, leaving out other studies on comparable issues written in other languages. Several studies studied a mixed population (T1DM and T2DM), with T2DM participants making up the majority, and the results were not uniformly diverse or consistent. However, the writers made use of their experience to analyze the data and clarify a broad idea despite the paucity of nursing literature. Due to the topic’s broadness, it was difficult to identify all connected topics, and Parnell et al.’s research frequently influenced our decisions [9]. In the current study, regions that needed more investigation emerged, especially regarding the T1DM population, which is still comparatively understudied in the literature. More precisely, the included studies span diverse geographic regions and age groups, each of which may present unique structural and contextual influences on health literacy. These population-level and contextual differences were not formally stratified in the synthesis, potentially limiting the comparability and generalizability of findings across settings.

Although some included studies involved caregivers, their health literacy and its impact on patient outcomes were not consistently assessed across the literature [36,47,54]. This represents a key gap and suggests the need for future studies examining caregiver–patient literacy dynamics, especially in pediatric and adolescent populations. Moreover, despite these limitations, this review helped to present a coherent and well-organized summary of the topic studied, paving the way for future research and posing direct implications in nursing and broader clinical practices. This review has facilitated the knowledge of the literature regarding HL in people with T1DM and assisted in identifying areas that require more attention in future research.

## 5. Conclusions

This scoping review underscores the pivotal role of HL in T1DM management and highlights the ongoing need for targeted research in this understudied population. Although some limitations affected the breadth and depth of evidence—such as the English-only focus and inclusion of mixed-population studies—this review offers a clear overview of current knowledge, identifies key gaps, and underscores the importance of integrating HL considerations into T1DM care. Researchers and clinicians could draw on these findings to develop targeted educational programs that align with patients’ literacy needs, incorporate standardized assessment tools to consistently measure HL, and embed HL-enhancing elements into both clinical guidelines and nursing practices. Such actionable steps will refine patient education, improve self-care behaviors, and foster stronger nurse–patient communication, ultimately paving the way for more personalized T1DM management and better health outcomes.

## Figures and Tables

**Figure 1 ijerph-22-00869-f001:**
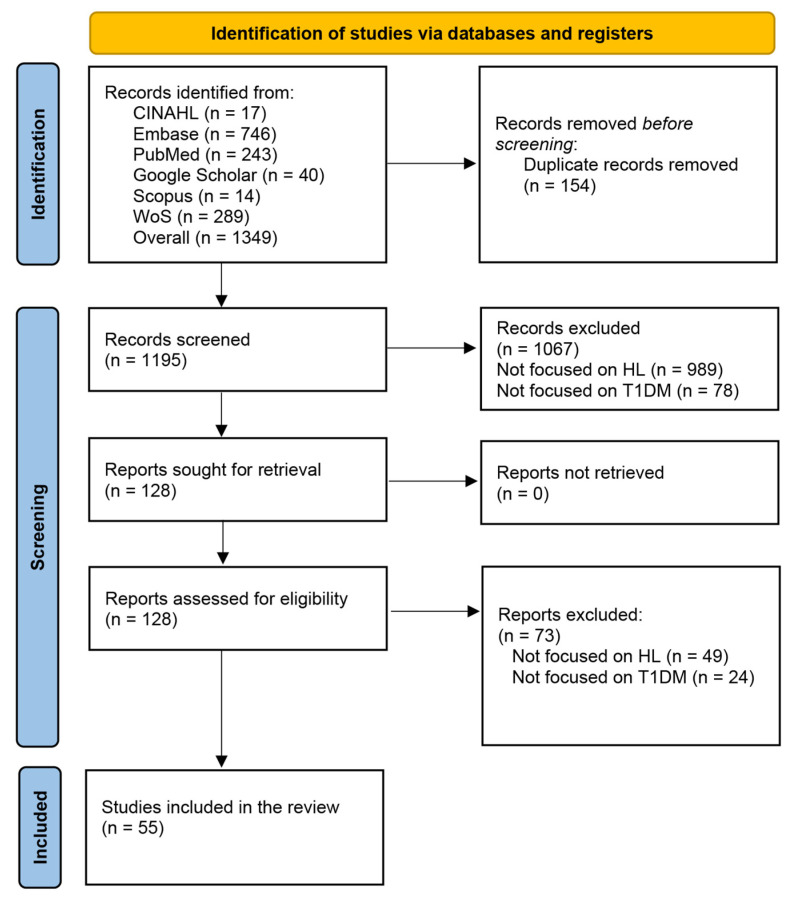
PRISMA 2020 flow diagram.

**Figure 2 ijerph-22-00869-f002:**
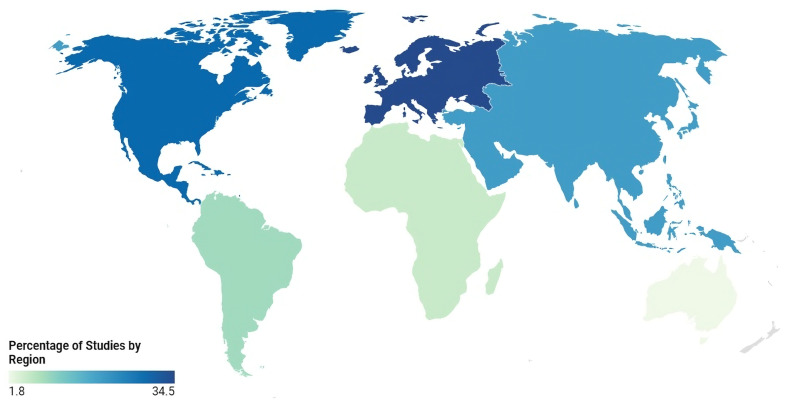
Heat map of study distribution across the globe.

**Figure 3 ijerph-22-00869-f003:**
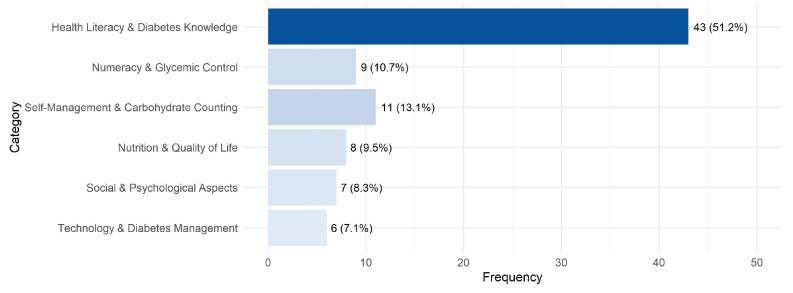
Distribution of grouped thematic areas.

**Figure 4 ijerph-22-00869-f004:**
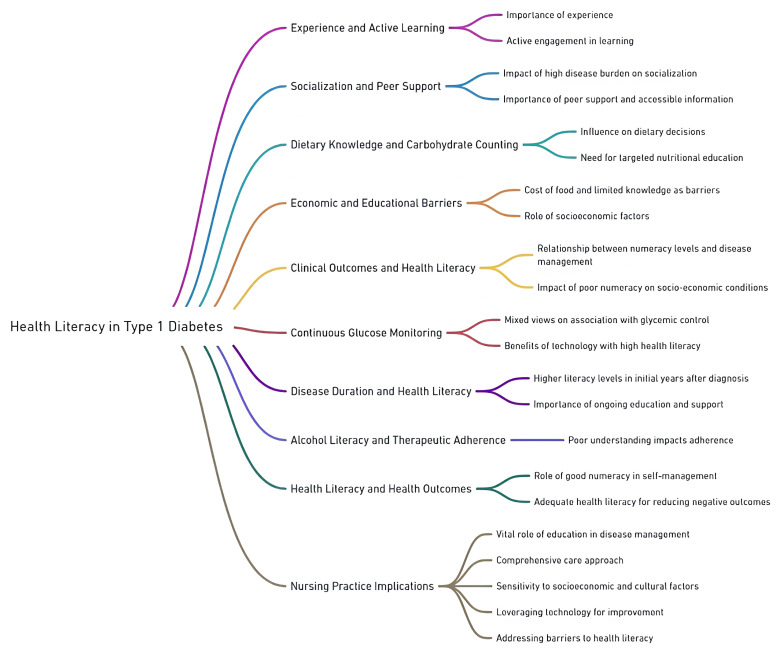
Mind map of factors influencing HL in people with T1DM and relationships between HL and health outcomes.

**Table 1 ijerph-22-00869-t001:** Characteristics of the included documents (*n* = 55).

		Count (%)	Reference(s)
**Years of Publication**			
	2005–2014	11 (29%)	[33,34,35,36,37,38,39,40,41,42]
	2015–2023	27 (71%)	[3,16,18,43,44,45,46,47,48,49,50,51,52,53,54,55,56,57,58,59,60,61,62,63,64,65,66,67,68]
**Geographic Region**			
	Africa	4 (10.5%)	[36,54,57,66]
	Asia	3 (7.9%)	[3,37,52]
	Europe	15 (39.5%)	[16,18,33,34,35,44,47,51,56,58,61,65,69,70,71]
	Middle East	3 (7.9%)	[48,49,60]
	North America	9 (23.7%)	[43,46,50,59,62,63,64,72,73]
	South America	4 (10.5%)	[45,53,67,74]
**Country economy**			
	Lower middle income	5 (13.1%)	[49,52,57,66,68]
	Upper middle income	6 (15.8%)	[36,45,53,60,74]
	High income	27 (71.1%)	[3,18,33,34,35,37,43,44,46,47,48,49,50,51,52,56,58,59,60,61,62,63,64,65,66,69,70,71,72,73]
**Journal discipline**			
	Nursing	1 (2.6%)	[54]
	Other	37 (97.4%)	[3,18,33,34,35,36,37,43,44,45,46,47,48,49,50,51,52,53,54,56,57,58,59,60,61,62,63,64,65,66,67,69,70,71,72,73,74]
**Type of publication**			
	Journal article	36 (94.7%)	[3,18,33,34,35,36,37,43,44,45,46,47,48,49,50,51,52,53,54,56,57,58,59,60,61,62,63,65,66,67,69,70,71,72,73,74]
	Other	2 (5.3%)	[50,64]
**Study design**			
	Literature review	4 (10.5%)	[3,59,63,65]
	Observational	28 (73.7%)	[33,34,35,36,37,43,44,46,47,48,49,50,51,52,53,56,59,60,61,62,64,65,66,67,69,70,71,73,74]
	Qualitative	4 (10.5%)	[33,54,57,58]
	Letters	1 (2.6%)	[50]
	Editorial	1 (2.6%)	[67]

**Table 2 ijerph-22-00869-t002:** Extracted themes.

Extracted Themes	References
Experience and Active Learning	[33,57]
Socialization and Peer Support	[43,56,58,66]
Dietary Knowledge and Carbohydrate Counting	[43,44,45,46,47,54,57]
Economic and Educational Barriers	[52,57]
Clinical Outcomes and Health Literacy	[3,35,36,45,47,48,49,50,53,56,59,62,63,64,65,67,69,71,72,73,74]
Continuous Glucose Monitoring	[16,47,58,61]
Disease Duration and Health Literacy	[18,51,57,66]
Alcohol Literacy and Therapeutic Adherence	[34,63,70]
Health Literacy and Health Outcomes	[3,57,59,60,64,65,72]
Nursing Practice Implications	[37,57,66]

## Data Availability

The data used in this study are available from the corresponding author upon reasonable request.

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
