# Peer review of "Health Literacy in People with Type 1 Diabetes: A Scoping Review"

_ijerph, 2025, doi:10.3390/ijerph22060869_

Round 1

Reviewer 1 Report

Comments and Suggestions for Authors

While the topic of health literacy (HL) in individuals with Type 1 Diabetes Mellitus (T1DM) is undoubtedly of clinical relevance, particularly in relation to self-management and outcomes, we found that the manuscript does not sufficiently advance the current body of knowledge in this area. The review reiterates well-established associations—such as the positive impact of HL on glycemic control and self-care—without offering substantial novel insights or a critical synthesis that might guide future interventions or policy.

Additionally, the methodological framework employed, while adequate for a scoping review, does not clearly articulate the criteria for inclusion, nor does it critically appraise the quality or heterogeneity of the included studies. This limits the interpretability and potential impact of the findings. We also noted that several of the conclusions drawn are broad and would benefit from more rigorous evidence and a clearer linkage to actionable clinical strategies.

Given this journal’s current focus on high-impact, original contributions or reviews that offer clear innovation in methodology, clinical translation, or theoretical framing, we believe this manuscript may be better suited to a specialized outlet with a narrower scope in diabetes education or public health literacy.

Author Response

Comment 1: While the topic of health literacy (HL) in individuals with Type 1 Diabetes Mellitus (T1DM) is undoubtedly of clinical relevance, particularly in relation to self-management and outcomes, we found that the manuscript does not sufficiently advance the current body of knowledge in this area. The review reiterates well-established associations—such as the positive impact of HL on glycemic control and self-care—without offering substantial novel insights or a critical synthesis that might guide future interventions or policy.

Response 1: We thank the Reviewer for this thoughtful comment and the opportunity to clarify the purpose and contribution of our work. This manuscript is framed as a scoping review, in line with the methodology outlined by Arksey & O’Malley and the JBI Manual for Evidence Synthesis. Rather than aiming to quantify associations or assess causality or other types of associations, as in a systematic review or meta-analysis, our objective was to map the existing literature on HL in people with T1DM, particularly in relation to thematic diversity, age-specific patterns, caregiver involvement, contextual disparities, and implications for intervention and nursing practice. The novelty of this work lies in its integrative synthesis across fragmented domains, as well as in its age-stratified analysis and global perspective, including underrepresented regions and caregiver insights. Such a review in T1DM is not available thus far. To address this concern and add clarity, we have now enhanced the Discussion to provide a clearer conceptual synthesis of mechanisms (see 4dr paragraph) and emphasized implications for practice and future research directions.

Comment 2: Additionally, the methodological framework employed, while adequate for a scoping review, does not clearly articulate the criteria for inclusion, nor does it critically appraise the quality or heterogeneity of the included studies. This limits the interpretability and potential impact of the findings. We also noted that several of the conclusions drawn are broad and would benefit from more rigorous evidence and a clearer linkage to actionable clinical strategies.

Response 2: We appreciate the Reviewer’s feedback and would like to clarify that this study adheres to the JBI methodology for scoping reviews, which does not require formal critical appraisal of included studies. As stated in Section 2.3 (Inclusion and Exclusion Criteria), we applied a broad, inclusive approach consistent with the aims of a scoping review (i.e., to map the extent and nature of available evidence on HL in T1DM rather than to evaluate effectiveness or synthesize outcomes. Specifically, we included studies addressing any dimension of HL in the context of T1DM management, while excluding those focused solely on T2DM or on psychometric validation unrelated to clinical content. To enhance clarity, we have refined the wording of the inclusion criteria in the Methods section and clarified the rationale for excluding psychometric tool validation studies to avoid overlap with COSMIN reviews. We also organized findings using structured frameworks—e.g., age-stratified analysis, conceptual domains, and geographic classifications—to manage heterogeneity and facilitate interpretation. While we agree that a critical synthesis of causality or effectiveness would require more rigorous study designs, such a task falls outside the scope of a scoping review and would require a separate systematic or meta-analytic effort.

Comment 3: Given this journal’s current focus on high-impact, original contributions or reviews that offer clear innovation in methodology, clinical translation, or theoretical framing, we believe this manuscript may be better suited to a specialized outlet with a narrower scope in diabetes education or public health literacy.

Response 3: We respectfully appreciate the Reviewer’s comment regarding journal fit and the focus on high-impact, original contributions. However, we believe that our scoping review offers a unique and valuable synthesis that aligns with the journal’s mission to inform public health practice, clinical research, and policy. This review presents a unique synthesis of 55 studies spanning diverse geographic regions, methodological approaches, and disciplinary perspectives, addressing the critical and timely public health issue of HL in the management of T1DM. Beyond mapping existing evidence, it highlights key thematic domains, identifies underexplored populations (such as caregivers and adolescents), and reveals conceptual and contextual gaps. These findings offer several valuable inputs for future research, such as designing targeted, age-appropriate HL interventions, and inform clinical applications, health policy planning, and interdisciplinary approaches to diabetes self-management support. The novelty of our contribution lies not only in the breadth of the evidence included but also in the thematic synthesis of HL domains (e.g., numeracy, nutrition, alcohol literacy, technology use), the identification of age-stratified patterns, and the inclusion of underrepresented populations and caregivers. These dimensions extend beyond the typical scope of narrowly focused diabetes education journals and offer insights relevant to public health, health promotion, chronic disease self-management, and health equity.

Reviewer 2 Report

Comments and Suggestions for Authors

Thank you for the opportunity to review this manuscript. This scoping review explores research questions related to health literacy and Type 1 Diabetes. The following are my comments:

-Line 65- 67. I would suggest adding references for the tools used.

-Line 97-100. The term “Type 1 Diabetes” does not appear in the research questions.

-Line 115. The database CINAHL is mentioned in Supplementary File 1 but not in the manuscript.

-Line 251. Please use a comma instead of a period for “1.349”. The same applies to “1.195” and “1.067”.

-Figure 1, I would recommend rephrasing “Wrong outcome” and “Wrong population”.

-Line 258. “… 55 studies met all criteria…” while Table 1 shows the “Characteristics of the included documents (n=38)”. Could you please clarify how the number changed from 55 to 38?

-Figure 3. I would suggest adding the count for each category.

-Line 559. Please double-check whether it should be “T1D” or “T1DM”.

Author Response

Comment 1: Thank you for the opportunity to review this manuscript. This scoping review explores research questions related to health literacy and Type 1 Diabetes.

Response 1: Thank you for having appreciated our work.

Comment 2: -Line 65- 67. I would suggest adding references for the tools used.

Response 2: We have added the references regarding the cited tools.

Comment 3: -Line 97-100. The term “Type 1 Diabetes” does not appear in the research questions.

Response 3: Thank you for this point. We have clarified the population (T1DM) in the research questions.

Comment 4: -Line 115. The database CINAHL is mentioned in Supplementary File 1 but not in the manuscript.

Response 4: Thank you for this point. We have corrected the main text, considering that CINAHL was one of the searched databases.

Comment 5: -Line 251. Please use a comma instead of a period for “1.349”. The same applies to “1.195” and “1.067”.

Response 5: Thank you for the clarification. We revised the text accordingly.

Comment 6: -Figure 1, I would recommend rephrasing “Wrong outcome” and “Wrong population”.

Response 6: We have revised Figure 1 according to your comment. We also found some typos, which were corrected in this version.

Comment 7: -Line 258. “… 55 studies met all criteria…” while Table 1 shows the “Characteristics of the included documents (n=38)”. Could you please clarify how the number changed from 55 to 38?

Response 7: Thank you for having detected this typo: "55 articles" is correct.

Comment 8: -Figure 3. I would suggest adding the count for each category.

Response 8: We agree with your comment and revised Figure 3 accordingly.

Comment 9: -Line 559. Please double-check whether it should be “T1D” or “T1DM”.

Response 9: Thank you for this point. We have checked the acronym and corrected an inconsistent use of T1D instead of T1DM.

Reviewer 3 Report

Comments and Suggestions for Authors

This scoping review overviewed the role of health literacy for the management of type 1 diabetes management by summarising the topic and themes of 55 studies. The authors suggested that in the literature, higher health literacy is associated improved elf-care, glycaemic control, and quality of life, especially the numeracy, nutrition literacy, digital health use etc. The review also stated the necessity for interventional studies based on health literacy. The topic of this review is relevant and the manuscript was well prepared. I have only some minor comments. 

1. For themes such as health literacy, the population of adolescents, young adults and older are different in scenarios. The discrepancy in geography should also be acknowledged or stratified. 
2. Caregivers and family members are also critical for people, especially children and adolescents, with type 1 diabetes. Was there any literature discussing their health literacy and outcomes, or their association with the literacy of the patients. 
3. The mechanism studies should be separately acknowledged in the review. For example, why health literacy plays a role in the management of type 1 diabetes? Is it through the improved understanding of the condition, or better use of new technologies? The review listed a number of detailed knowledges, but it is interesting to find the abstracted concept behind them. 

Author Response

Comment 1: This scoping review overviewed the role of health literacy for the management of type 1 diabetes management by summarising the topic and themes of 55 studies. The authors suggested that in the literature, higher health literacy is associated improved elf-care, glycaemic control, and quality of life, especially the numeracy, nutrition literacy, digital health use etc. The review also stated the necessity for interventional studies based on health literacy. The topic of this review is relevant and the manuscript was well prepared. I have only some minor comments. 

Response1: Thank you a lot for having appreciated our work.

Comment 2: For themes such as health literacy, the population of adolescents, young adults and older are different in scenarios. The discrepancy in geography should also be acknowledged or stratified. 

Response2: We thank the Reviewer for this important observation. In response, we revised the Discussion section to acknowledge the heterogeneous scenarios across age groups and geographic settings, noting how differences in developmental stages, health system challenges, and socio-educational contexts may influence health literacy and diabetes management. Additionally, we explicitly addressed this issue in the Limitations, clarifying that although studies included diverse populations and contexts, these variations were not formally stratified in the synthesis, which may limit the comparability and generalizability of the findings. These clarifications have been added to ensure transparency and to guide interpretation of the results.

Comment 3: Caregivers and family members are also critical for people, especially children and adolescents, with type 1 diabetes. Was there any literature discussing their health literacy and outcomes, or their association with the literacy of the patients. 

Response 3: We thank the Reviewer for this insightful observation. A limited number of studies included in our review did address the role of caregivers and family members, particularly in pediatric and adolescent T1DM populations. For example, studies by Moosa et al. (2011), Owusu et al. (2023), and Lappenschaar et al. (2012) acknowledged caregiver involvement and highlighted that numeracy and diabetes-related knowledge among caregivers may influence patient self-management and glycemic control. These findings have now been incorporated into the results (previously they were mentioned only in supplementary file 2), and we have also noted in the Limitations that caregiver-specific HL was not a primary focus across most studies, highlighting a gap for future research.

Comment 4: The mechanism studies should be separately acknowledged in the review. For example, why health literacy plays a role in the management of type 1 diabetes? Is it through the improved understanding of the condition, or better use of new technologies? The review listed a number of detailed knowledges, but it is interesting to find the abstracted concept behind them. 

Response 4: We thank the Reviewer for highlighting the importance of identifying the mechanisms through which HL influences disease management. In response, we have revised the Discussion to include an integrative paragraph that outlines key pathways observed across the literature. These include enhanced cognitive understanding, improved interaction with diabetes-related technologies, strengthened psychological competencies such as self-efficacy, and modulation by structural factors like socioeconomic status. This addition provides a synthesized conceptual framework to better explain how HL supports self-management in T1DM and aligns with the broader goal of deriving actionable insights from the literature.